# Comparison of Self-Administered Web-Based and Interviewer Printed Food Frequency Questionnaires for Dietary Assessment in Italian Adolescents

**DOI:** 10.3390/ijerph16111949

**Published:** 2019-06-01

**Authors:** Martina Barchitta, Andrea Maugeri, Ottavia Agrifoglio, Giuliana Favara, Claudia La Mastra, Maria Clara La Rosa, Roberta Magnano San Lio, Antonella Agodi

**Affiliations:** Department of Medical and Surgical Sciences and Advanced Technologies “GF Ingrassia”, University of Catania, 95123 Catania, Italy; martina.barchitta@unict.it (M.B.); andreamaugeri88@gmail.com (A.M.); ottavia.agrifoglio@gmail.com (O.A.); giuliana.favara@gmail.com (G.F.); claudia.lamastra@libero.it (C.L.M.); mariclalarosa@gmail.com (M.C.L.R.); robimagnano@gmail.com (R.M.S.L.)

**Keywords:** dietary assessment, epidemiology, nutrition, public health, adolescence, foods, nutrients

## Abstract

Innovative tools for assessing food and nutrient intakes in adolescence are essential to uncover the long-term effects of diet on chronic diseases. Here, we developed and compared a web-based self-administered food frequency questionnaire (web-FFQ) with a traditional interviewer printed FFQ (print-FFQ) among 174 Italian adolescents (aged 15–18 years). To assess the reliability of the web-FFQ compared with the print-FFQ, we used Spearman’s rank correlation coefficients, Wilcoxon rank test, quartile misclassification analysis, Cohen’s weighted kappa and the Bland–Altman method. Correlation coefficients ranged from 0.14 (i.e., pizza) to 0.67 (i.e., raw vegetables) for foods, and from 0.45 (i.e., monounsaturated fatty acids, MUFA) to 0.62 (i.e., zinc) for nutrients. Results from the Wilcoxon rank test showed that food and nutrient intakes were comparable between two FFQs, except for nuts, shellfish, fruit juices and MUFA. Adolescents classified into the same or adjacent quartiles ranged from 68.2% (i.e., tea) to 89.1% (i.e., raw vegetables and dipping sauces) for foods, and from 77.2% (i.e., vitamin C) to 87.2% (i.e., folate and calcium) for nutrients. Except for pizza, the weighted kappa indicated moderate to substantial agreement for other foods and nutrients. Finally, we demonstrated that the web-FFQ significantly overestimated shellfish and fruit juice intakes, while it underestimated nuts, canned fish, olive oil, total energy intake, fatty acids and calcium. The limits of agreement analysis indicated moderate to wide individual differences for all groups. In conclusion, our self-administered web-FFQ represents an easy, suitable and cost-effective tool for assessing food and nutrient intakes in adolescents. However, the wide individual differences in level of agreement suggest that additional refinements and calibrations are necessary to investigate the effects of absolute nutrient intakes at the individual level.

## 1. Introduction

Although adolescents—young people between 10 and 19 years old—are often considered a healthy group, most of them suffer chronic ill health or disability [1]. Several population-based studies from Western countries showed that 20–30% of adolescents had a chronic illness [2]. In the last decades, global reduction in the burden from infectious disease has been accompanied by a growing importance of non-communicable diseases (NCDs), such as mental and substance use disorders, cancer, congenital anomalies and hemoglobinopathies [1]. However, the leading causes of death and disability vary with age, sex and socioeconomic status, so challenges for each country may be different [1]. In addition, several NCDs in adulthood have their roots in childhood and adolescence [3]. Indeed, behaviors underpinning the major NCDs—tobacco and alcohol use, unhealthy diet and physical inactivity, overweight and obesity—usually start or are reinforced during the second decade of life [4,5,6]. Due to the rising number of obese infants, children and adolescents, the World Health Organization set up the Commission on Ending Childhood Obesity, which in turn proposed a set of recommendations to tackle the childhood and adolescent obesity epidemic [7]. This Commission recognized that a multidisciplinary approach is crucial for sustained progress. For instance, countries should survey prevalence and trends in childhood and adolescent obesity at national, regional and local levels. They should also collect data on diet and physical activity of children and adolescents across different socioeconomic groups and settings. Although some data are collected, a significant lack that needs to be filled remains for children over 5 years and adolescents [8]. This data might guide the development of appropriate priorities and strategies, providing a benchmark against which the efficacy of policies and programs can be measured [8]. Large-scale epidemiologic studies and surveys often use food frequency questionnaires (FFQs) to assess food and nutrient intakes over an extended period of time. Although food diaries and 24 h dietary recalls are considered more precise than FFQs [9,10], it has been demonstrated that the latter correctly rank individuals according to their dietary intakes [11]. While different FFQs have been validated in adolescents [12,13,14,15,16], the need of innovative methods, such as a web-based FFQ, has been recently highlighted for uncovering the long-term effects of diet on chronic diseases [17]. In 2014, researchers from the Adolescents and Surveillance System for Obesity prevention (ASSO) project reported on the relative reproducibility of a web-based FFQ among Italian adolescents [17]. Despite this progress, more efforts are needed to develop web-based tools that are easier, more suitable and more cost-effective than traditional methods for assessing food and nutrient intakes [17]. To fill this gap, in the present study we aim to compare a self-administered web-based FFQ (web-FFQ) with an interviewer printed FFQ (print-FFQ) for dietary assessment in Italian adolescents aged 15 to 18 years.

## 2. Materials and Methods

### 2.1. Study Design and Data Collection

The present study included adolescents (aged 15 to 18 years), attending three high schools in the urban area of Eastern Sicily. Ethical approval for the study protocol was given by the school boards of involved institutions, and informed consent was signed by parents or by the adolescents themselves if they were 18 years old or older. More details on protocols and characteristics of study population were reported elsewhere [18]. During classroom time and under the supervision of trained teachers, all the adolescents were invited to participate in this comparative study and were provided with information sheets. We used data from adolescents who completed both FFQs with plausible total energy intake [19]. A structured ad hoc questionnaire was administered to collect data on lifestyles (i.e., physical activity and smoking status) and sociodemographic factors of parents. Physical activity was assessed using the International Physical Activity Questionnaire for adolescents (IPAQ-A) [20]. Self-reported body weight and height were used to calculate body mass index (BMI) according to the World Health Organization criteria for adolescents [21].

### 2.2. Dietary Assessment

We first adapted the interviewer print-FFQ and the self-administered web-FFQ from a 95-item interviewer print-FFQ, which was previously used in the research on the effect of dietary habits on women’s health [22,23,24,25]. The latter, in turn, was updated from a 46-item FFQ validated for the assessment of folate intake in Italian women of child-bearing age [26]. The print-FFQ has already been used to assess the association between dietary patterns and school performance in the present study population of adolescents [18].

Both FFQs included questions on 95 food items categorized into 39 food groups and eight food categories. In particular, data on frequency of consumption (from “never” to “more than twice per day”) and the portion size (small, half a medium portion size; medium; large, 1.5 times or more than the medium portion size) for each food item were collected. Portion sizes were assessed by using images extracted from a photographic atlas. First, adolescents were invited to fill the web-FFQ following information sheets and instructions by trained epidemiologists. Next, the print-FFQ was administered by trained interviewers within the week after the web-FFQ administration. From both FFQs, daily food intakes were calculated by multiplying the daily frequency of consumption with the portion size of each food group. Daily nutrient intakes were calculated using the table of food composition of the US Department of Agriculture (http://ndb.nal.usda.gov/). Prior to further analysis, nutrient and food intakes were adjusted for total energy intake using the residual method [27].

### 2.3. Statistical Analysis

Prior to analysis, we excluded print-FFQs and web-FFQs in which the total energy intake per day was implausible (total energy intake <700 kcal or >4500 kcal), as previously recommended by Rockett et al. [19]. Moreover, the normal distribution of variables was checked using the Kolmogorov–Smirnov test. Adolescent characteristics were reported as proportions, mean or standard deviation (SD), or median and interquartile range (IQR), and compared between sexes using the Chi-squared test for categorical variables or the Mann–Whitney U test for skewed continuous variables.

Several methods were used to assess the reliability of the web-FFQ for the assessment of food and nutrient intakes, in comparison with the print-FFQ. Although several lines of evidence suggested that the correlation coefficient might be a misleading indicator of agreement [28,29,30], we first calculated Spearman’s rank correlation coefficients to allow comparisons with previous studies that did not use the other methods. Wilcoxon rank test was used to compare differences between the two FFQs. The quartile misclassification analysis and Cohen’s weighted kappa were applied to assess the correct ranking ability of the web-FFQ compared with the print-FFQ. According to Masson and colleagues, at least 50% of subjects should be classified into the same quartile, no more than 10% should be classified into the opposite quartile and Cohen’s weighted kappa should be above 0.4 [31]. The Bland–Altman method was used to assess the agreement between absolute food and nutrient intakes estimated by each FFQ [32]. This method also helps identify the direction and consistency of any bias across the range of intakes [32]. Prior to analysis, food and nutrient intakes were log-transformed due to their skewed distributions (i.e., a constant value of 0.01 was added to food or nutrient intakes equal to zero). For each food group and nutrient, mean agreement was calculated as the average of all individual differences between FFQs, according to the following formula: (∑log(web-FFQ)-log(print-FFQ))/n. The limits of agreement (LOA) were calculated as mean agreement ± 1.96 * (standard deviation of differences between methods). Mean agreement indicates how well the web-FFQ and print-FFQ agree on average, while the LOA designate the interval within which 95% of all individual differences between FFQs are expected to fall [32]. Finally, we plotted individual differences in intakes between methods (log(web-FFQ–log(print-FFQ)) against their average. To evaluate whether agreement between the web-FFQ and print-FFQ was constant across the range of intakes, we estimated the regression slope (β) of the average of the two methods on their differences [32]. Statistical analyses were performed using the SPSS software (version 22.0, SPSS, Chicago, IL, USA) and *p*-values less than 0.05 were considered statistically significant.

## 3. Results

A total of 179 adolescents, aged 15 to 18 years (median = 16 years), completed both FFQs, out of which five were excluded due to implausible energy intakes. Thus, 174 adolescents were included in this comparative study (median age =16 years; IQR = 2). No significant differences between included and excluded adolescents were evident. Mean BMI was 20.4 kg/m^2^ (SD = 3.1 kg/m^2^), with boys (47.8%) reporting higher weight, height and BMI than girls (*p*-values < 0.001). According to BMI, 91% of adolescents were normal weight (88.2% boys and 93.5% girls) and only 1.7% were obese (2.4% boys and 1.1% girls). Moreover, boys spent more time in moderate-to-vigorous physical activity than girls (*p* < 0.001). No differences in sociodemographic characteristics of parents and smoking status were evident between sex.

Table 1 displays Spearman’s correlation coefficients for food groups that ranged from 0.14 (i.e., pizza) to 0.67 (i.e., raw vegetables). Results from the Wilcoxon rank test indicated that food intakes were comparable between the two FFQs, except for nuts, shellfish and fruit juices. The percentage of subjects classified into the same or adjacent quartiles by the two FFQs ranged from 68.2% (i.e., tea) to 89.1% (i.e., raw vegetables and dipping sauces). In contrast, the web-FFQ misclassified from 0% (i.e., dipping sauces) to 13.1% (i.e., fish) of subjects into the opposite quartile compared to the print-FFQ. The weighted kappa values indicated substantial agreement for raw vegetables, yoghurt, eggs, coffee, dipping sauces and soup (weighted kappa from 0.6 to 0.8), and low agreement for pizza (weighted kappa below 0.2).

As an example, Figure 1 shows Bland–Altman plots of the average between the web-FFQ and print-FFQ versus their difference, the mean agreement, LOA and regression slope for raw vegetables, fruit, red meat and white bread. The mean agreement between the web-FFQ and print-FFQ was significantly different from zero for five food groups: on one hand, the web-FFQ significantly underestimated nuts, canned fish and olive oil intakes; on the other hand, it overestimated shellfish and fruit juice intakes. However, the LOA indicated wide individual differences between the web-FFQ and the print-FFQ for all food groups. In particular, the mean agreement between the web-FFQ and print-FFQ was significantly poorer at high levels of intake (i.e., significant positive regression slope) for offal, shellfish and fruit salad (*β* = 0.514, 0.258 and 0.170, respectively) (Table 2).

Table 3 displays Spearman’s correlation coefficients that ranged from 0.45 (i.e., monounsaturated fatty acids, MUFA) to 0.62 (i.e., zinc). Results from the Wilcoxon rank test indicated that nutrient intakes were comparable between the two FFQs, except for MUFA. The percentage of subjects classified into the same or adjacent quartiles by the two FFQs ranged from 77.2% (i.e., vitamin C) to 87.2% (i.e., folate and calcium). In contrast, the web-FFQ misclassified from 1.7% (i.e., zinc) to 13.1% (i.e., MUFA) of subjects into the opposite quartile compared to the print-FFQ. The weighted kappa values indicated moderate agreement for all nutrients (weighted kappa from 0.4 to 0.6).

As an example, Figure 2 shows Bland–Altman plots of the average between the web-FFQ and print-FFQ versus their difference, the mean agreement, LOA and regression slope for total energy intake, saturated fatty acids (SFA), MUFA and folate. The mean agreement between the web-FFQ and print-FFQ was significantly different from zero for total energy intake, SFA, MUFA, polyunsaturated fatty acids (PUFA) and calcium, which were underestimated by the web-FFQ. The LOA indicated moderate to wide individual differences between the web-FFQ and the print-FFQ for all nutrients (Table 2). In particular, the mean agreement between the web-FFQ and print-FFQ was significantly poorer at high levels of intake for SFA, (*β* = 0.179); on the contrary, the difference in MUFA intake was lower at lower levels of intake (*β* = −0.427) (Table 4).

## 4. Discussion

Despite the need for innovative tools for gathering dietary data in childhood and adolescence, few studies have developed and validated web-based tools for dietary assessment. To the best of our knowledge, only a web-based FFQ for dietary assessment in Italian adolescents has been developed thus far [17] and validated [33] within the Adolescents and Surveillance System for the Obesity prevention (ASSO) project.

In the current study, we adapted a self-administered web-FFQ from a validated interviewer-administered print-FFQ, which used the previous month as a reference period [23,24]. Next, we reported on correlations, ranking ability and mean agreement between food and nutrient intakes in Italian adolescents, estimated by the self-administered web-FFQ in comparison with the interviewer print-FFQ. To allow comparison with previously reported tools, we first assessed Spearman’s rank correlation coefficients between the web-FFQ and print-FFQ. Except for pizza, correlation coefficients of food and nutrient intakes were in line with previous studies using a dietary reference method. As reviewed by Cade and colleagues and Tabacchi and colleagues, a good correlation between two methods is generally considered for a coefficient value of 0.4 [34,35]. Specifically, the correlation coefficients ranged from 0.14 (i.e., pizza) to 0.67 (i.e., raw vegetables) for foods, and from 0.45 (i.e., MUFA) to 0.62 (i.e., zinc) for nutrients. Results from the Wilcoxon rank test showed that food and nutrient intakes were comparable between the two FFQs, except for nuts, shellfish, fruit juices and MUFA. Moreover, the ranking ability of our web-FFQ was adequate, as demonstrated by quartile misclassification analysis and Cohen’s weighted kappa. In fact, the percentage of subjects classified into the same or adjacent quartiles ranged from 68.2% (i.e., tea) to 89.1% (i.e., raw vegetables and dipping sauces) for foods, and from 77.2% (i.e., vitamin C) to 87.2% (i.e., folate and calcium). These results were in line with those reported by previous studies using self-administered traditional or web-based FFQs [17,33,36,37]. According to Masson and colleagues [31], misclassification was higher for fish and MUFA, with 13.1% of subjects classified into the opposite quartile. For food groups, the weighted kappa values indicated substantial agreement for raw vegetables, yoghurt, eggs, coffee, dipping sauces and soup, moderate agreement for other foods and low agreement for pizza. For all nutrients, the weighted kappa values indicated moderate agreement. Finally, we used the Bland–Altman method, which assesses the agreement between absolute intakes estimated by each FFQ [32]. We demonstrated that the mean agreement between FFQs was significantly different from zero for five foods and five nutrients: while the web-FFQ significantly overestimated shellfish and fruit juice intakes, it underestimated nuts, canned fish, olive oil, total energy intake, SFA, MUFA, PUFA and calcium. Although a good distribution of values around the mean difference was reported in general, the LOA analysis indicated moderate to wide individual differences for all food and nutrient groups [32]. This was in line with previous validation studies of web-based FFQs [33,38,39,40]. In particular, the mean agreement between the web-FFQ and print-FFQ was significantly poorer at high levels of intake for offal, shellfish, fruit salad and SFA, while it was lower at lower levels of MUFA intake.

We acknowledge the limitations of our study. Since FFQs rely on memory, the web-FFQ and print-FFQ share measurement errors [35]. Weighed record or food record are the first methods for validating FFQs, followed by 24 h recalls. However, similar to FFQs, they are prone to a degree of misreporting [35]. Accordingly, Cade and colleagues reviewed dietary assessment tools which were used as a reference measure. They showed that almost half the validation studies used weighed record or food record, about a fifth used 24 h recall and more than one-tenth used another FFQ [34]. Potential measurement errors might partially explain the low total energy intake observed among our study population. Indeed, we reported median values of ~1900 kcal, which are lower than those reported by a previous study [17]. However, the IQR of energy intake assessed by the web-FFQ was 1328–2618, similar to that reported by Tabacchi et al. using a weighted food record [33]. Since we cannot completely exclude misleading estimates, we only included adolescents with plausible total energy intake, and adjusted nutrient and food intakes for total energy intake. We considered implausible total energy intake as <700 kcal or >4500 kcal according to a previous study [19]. Rockett and colleagues defined outliers or implausible responses to the questionnaire as energy intake less than 500 kcal per day or greater than 5000 kcal per day [19]. In our study we used more stringent values to reduce the risk of potential measurement errors. The present study also has several strengths, including the number of respondents, which exceeds most other reliability studies. Our sample size of 175 adolescents allowed correlations, ranking ability and Bland–Altman analyses, which require a sample size of at least 100 subjects [34]. Taking into account its strengths and weaknesses, in our opinion, the proposed web-FFQ could be used for gathering dietary data in adolescence in future large-scale research against the increasing burden of NCDs.

## 5. Conclusions

In conclusion, our self-administered web-FFQ represents an easy, suitable and cost-effective tool for assessing food and nutrient intakes in adolescents. Although mean agreement between absolute intakes was less than ideal, the web-FFQ showed moderate to substantial ability to rank adolescents into the same or adjacent quartile as the interviewer print-FFQ. However, the wide individual differences in the level of agreement for all food and nutrient groups suggest that additional refinements and calibrations are needed to investigate the effects of absolute nutrient intakes at the individual level.

## Figures and Tables

**Figure 1 ijerph-16-01949-f001:**
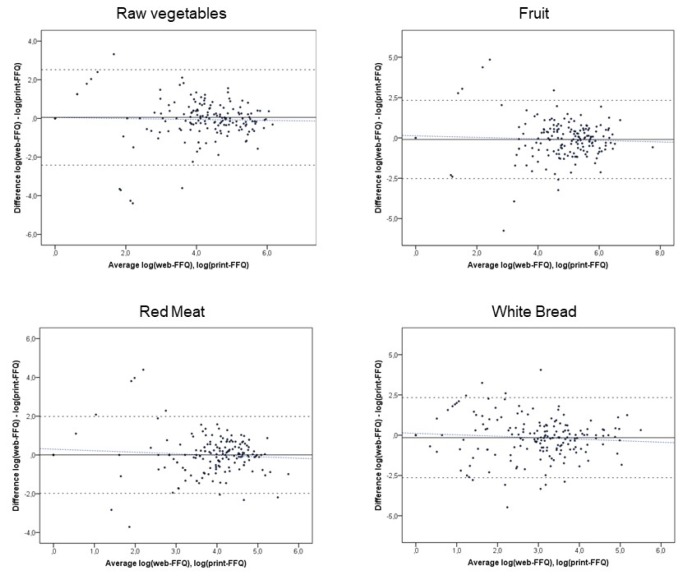
Bland-Altman plots of individual differences in food intakes (raw vegetables, fruit, red meat and white bread) between the web-FFQ and print-FFQ against their averages; the upper and lower 95% limits of agreement (dashed lines), mean agreement (solid line) and regression slope (blue dashed line) are indicated.

**Figure 2 ijerph-16-01949-f002:**
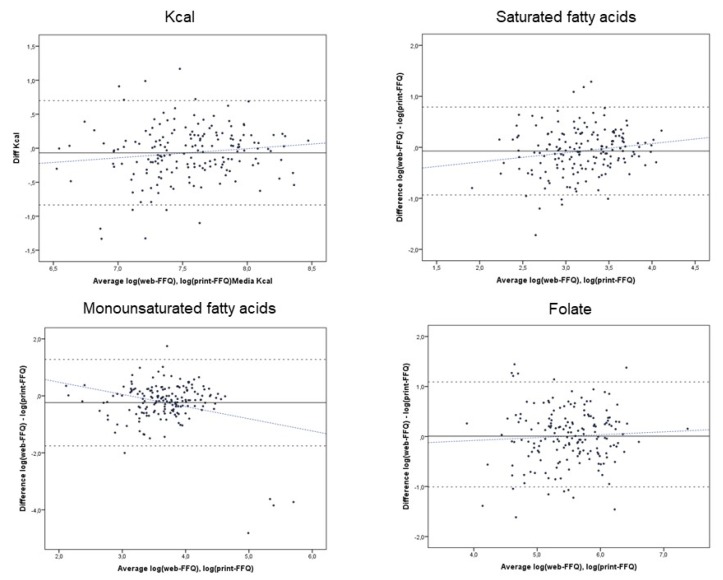
Bland–Altman plots of individual differences in nutrient intakes (total energy intake, saturated fatty acids, monounsaturated fatty acids and folate) between the web-FFQ and print-FFQ against their averages; the upper and lower 95% limits of agreement (dashed lines), mean agreement (solid line) and regression slope (blue dashed line) are indicated.

**Table 1 ijerph-16-01949-t001:** Daily food intakes, Spearman’s correlation coefficients and quartile misclassification analysis between the web-FFQ and print-FFQ.

Food Groups (g)	Web-FFQ			Print-FFQ			Wilcoxon Rank Test	Classified in the Same Quartile	Classified in the Adjacent Quartile	Classified in the Opposite Quartile	Weighted Kappa	Spearman’s Correlation Coefficient
	Median	Q1	Q3	Median	Q1	Q3						
Potatoes	5.0	0.0	20.0	7.5	0.0	15.0	0.872	37.7%	42.3%	4.6%	0.429	0.46
Cooked vegetables	76.0	43.3	124.7	80.0	46.5	123.3	0.799	46.9%	39.4%	1.7%	0.587	0.60
Raw vegetables	83.3	36.7	135.0	77.5	33.3	163.3	0.518	57.1%	32.0%	2.3%	0.654	0.67
Legumes	21.3	8.7	45.8	22.7	11.8	44.0	0.283	49.7%	36.0%	2.3%	0.576	0.56
Fruit	149.3	74.7	263.0	176.7	79.3	285.3	0.206	39.4%	45.1%	4.0%	0.490	0.55
Nuts	0.5	0.0	2.0	1.0	0.0	5.0	0.002	48.6%	31.4%	9.1%	0.471	0.50
Milk	50.0	0.0	200.0	50.0	0.0	200.0	0.544	54.9%	32.6%	4.6%	0.601	0.65
Yoghurt	0.0	0.0	16.7	0.0	0.0	16.7	0.234	-	-	-	-	0.65
Curd cheese	16.7	8.3	33.3	16.7	8.3	33.3	0.316	46.9%	28.0%	6.9%	0.370	0.38
Ripened cheese	13.3	3.3	30.0	16.7	3.3	33.3	0.172	50.3%	33.7%	2.9%	0.560	0.61
Pasta	60.0	30.7	70.0	56.0	32.7	77.2	0.195	44.6%	34.9%	2.3%	0.478	0.49
Rice	6.7	1.2	10.7	5.3	1.2	11.0	0.539	46.9%	34.3%	4.6%	0.474	0.48
White bread	20.0	9.8	36.1	23.2	9.0	45.5	0.054	44.6%	34.3%	4.6%	0.431	0.48
Whole wheat bread	0.4	0.0	16.1	0.0	0.0	17.7	0.497	-	-	-	-	0.50
Breakfast cereals	3.0	0.0	8.0	2.0	0.0	12.0	0.241	52.6%	30.9%	4.0%	0.563	0.62
Red meat	56.0	32.0	96.0	54.0	32.0	82.0	0.611	45.1%	37.7%	4.6%	0.506	0.56
White meat	24.0	8.0	48.0	24.0	8.0	48.0	0.808	40.6%	34.3%	4.6%	0.279	0.31
Processed meat	34.3	16.0	55.3	33.2	17.8	58.7	0.555	47.4%	37.1%	1.7%	0.568	0.59
Offal	0.0	0.0	0.0	0.0	0.0	0.0	0.147	-	-	-	-	0.49
Canned fish	6.9	0.0	13.9	6.9	0.0	13.9	0.550	51.4%	32.6%	2.9%	0.523	0.54
Shellfish	5.0	0.0	20.0	5.0	0.0	15.0	<0.001	47.4%	27.4%	8.6%	0.406	0.42
Fish	24.7	11.8	54.3	26.0	10.0	45.0	0.641	46.9%	38.9%	13.1%	0.591	0.62
Eggs	16.0	8.0	32.0	16.0	8.0	32.0	0.111	49.1%	38.3%	1.1%	0.622	0.62
Vegetable oil	0.0	0.0	4.0	0.0	0.0	4.0	0.277	-	-	-	-	0.38
Olive oil	30.0	15.0	36.0	30.0	15.0	36.0	0.519	42.9%	44.6%	1.7%	0.337	0.39
Butter and margarine	0.7	0.0	2.7	0.7	0.0	1.3	0.054	56.0%	25.1%	5.7%	0.552	0.56
Sweets and refined sugar	21.2	10.5	37.3	18.0	10.5	36.0	0.141	47.4%	35.4%	4.0%	0.499	0.55
Fruit juice	33.3	0.0	133.3	13.3	0.0	80.0	<0.001	45.7%	34.3%	5.7%	0.479	0.54
Coffee	5.0	0.0	50.0	5.0	0.0	40.0	0.132	55.4%	31.4%	3.4%	0.605	0.65
Tea	4.0	0.0	33.3	4.0	0.0	53.3	0.067	55.6%	12.6%	8.0%	0.451	0.50
Dipping sauces	1.3	0.0	2.7	1.3	0.0	2.7	0.070	53.1%	36.0%	0.0%	0.605	0.61
Soup	7.5	0.0	20.0	7.5	0.0	20.0	0.304	57.1%	28.0%	2.3%	0.624	0.64
Pizza	30.0	12.0	40.0	30.0	20.0	40.0	0.141	37.7%	37.1%	6.3%	0.154	0.14
Wine	0.0	0.0	0.0	0.0	0.0	0.0	0.940	-	-	-	-	0.48
Alcoholic drink	0.0	0.0	4.0	0.0	0.0	4.0	0.826	-	-	-	-	0.55
Beer	0.0	0.0	33.0	0.0	0.0	22.0	0.377	-	-	-	-	0.63
Salty snacks	2.2	0.4	4.3	2.2	0.8	5.0	0.409	39.4%	40.6%	4.6%	0.446	0.51
Fries	18.0	6.0	24.0	12.0	6.0	24.0	0.681	45.1%	36.0%	3.4%	0.464	0.48
Fruit salad	0.0	0.0	10.0	0.0	0.0	5.0	0.058	-	-	-	-	0.54

Abbreviations: FFQ, food frequency questionnaire; Q1, first quartile; Q3, third quartile.

**Table 2 ijerph-16-01949-t002:** Daily nutrient intakes, Spearman’s correlation coefficients and quartile misclassification analysis between the web-FFQ and print-FFQ.

Nutrients	Web-FFQ			Print-FFQ			Wilcoxon Rank Test	Classified in the Same Quartile	Classified in the Adjacent Quartile	Classified in the Opposite Quartile	Weighted Kappa	Spearman’s Correlation Coefficient
	Median	Q1	Q3	Median	Q1	Q3						
Total energy intake, kcal	1888.2	1328.6	2618.4	1926.9	1503.0	2583.3	0.092	42.5%	43.0%	2.8%	0.538	0.58
SFA, mg	22.6	15.6	34.4	24.2	17.8	32.7	0.129	44.1%	39.7%	2.8%	0.524	0.57
MUFA, mg	34.4	24.8	53.2	42.3	29.6	57.6	<0.001	38.0%	41.9%	5.0%	0.410	0.45
PUFA, mg	12.9	8.6	17.9	13.8	9.7	18.6	0.088	43.6%	38.5%	3.9%	0.474	0.50
Folate, µg	252.2	163.6	385.8	251.9	162.4	376.5	0.422	40.8%	46.4%	2.2%	0.564	0.61
Iron, mg	12.7	8.9	16.8	12.6	9.2	19.0	0.209	43.6%	40.2%	3.4%	0.513	0.58
Calcium, mg	776.1	558.9	1196.2	812.2	584.5	1207.4	0.128	47.5%	39.7%	2.8%	0.578	0.59
Magnesium, mg	261.1	180.9	342.1	251.5	188.6	357.2	0.826	39.7%	44.1%	2.8%	0.506	0.56
Zinc, mg	9.5	6.5	13.6	9.4	6.8	13.4	0.334	46.9%	40.2%	1.7%	0.596	0.62
Vitamin A, µg	736.9	472.3	1186.0	793.6	501.0	1185.5	0.621	45.3%	40.2%	2.2%	0.560	0.59
Vitamin B1, mg	1.4	1.0	1.9	1.4	1.0	2.0	0.467	39.7%	42.5%	2.8%	0.499	0.52
Vitamin B6, mg	2.0	1.4	2.9	2.1	1.5	2.9	0.525	43.6%	39.1%	3.4%	0.484	0.53
Vitamin C, mg	119.2	50.4	184.4	86.3	48.3	159.9	0.055	42.5%	35.2%	3.9%	0.421	0.47
Vitamin D, µg	5.3	3.0	9.6	5.4	3.3	9.5	0.550	44.7%	37.4%	4.5%	0.475	0.50

Abbreviations: FFQ, food frequency questionnaire; Q1, first quartile; Q3, third quartile; SFA, saturated fatty acids; MUFA, monounsaturated fatty acids; PUFA, polyunsaturated fatty acids.

**Table 3 ijerph-16-01949-t003:** Mean agreement and limits of agreement between food intakes estimated by the web-FFQ and the print-FFQ.

Food Groups	Mean Agreement (95% CI)	Lower LOA	Upper LOA	Slope (*β*)
Potatoes	−0.030 (−0.254, 0.194)	−2.973	2.913	0.054
Cooked vegetables	−0.089 (−0.220, 0.042)	−1.809	1.631	0.111
Raw vegetables	0.050 (−0.138, 0.238)	−2.421	2.521	−0.126
Legumes	0.038 (−0.146, 0.222)	−2.384	2.460	0.017
Fruit	−0.097 (−0.281, 0.088)	−2.520	2.327	−0.047
Nuts	−0.220 (−0.361, −0.079)	−2.072	1.632	−0.139
Milk	0.151 (−0.162, 0.465)	−3.966	4.269	−0.073
Yoghurt	−0.195 (−0.414, 0.023)	−3.066	2.675	−0.065
Curd cheese	0.065 (−0.159, 0.288)	−2.869	2.999	0.032
Ripened cheese	−0.042 (−0.222, 0.139)	−2.421	2.337	−0.102
Pasta	−0.087 (−0.235, 0.060)	−2.029	1.854	−0.009
Rice	−0.052 (−0.243, 0.140)	−2.567	2.464	0.009
White bread	−0.153 (−0.341, 0.036)	−2.638	2.332	−0.092
Whole wheat bread	−0.086 (−0.352, 0.181)	−3.586	3.414	−0.088
Breakfast cereals	−0.040 (−0.214, 0.135)	−2.334	2.254	−0.097
Red meat	0.002 (−0.149, 0.153)	−1.978	1.983	−0.078
White meat	−0.040 (−0.294, 0.215)	−3.385	3.306	0.010
Processed meat	0.030 (−0.089, 0.149)	−1.536	1.596	−0.090
Offal	0.020 (−0.052, 0.093)	−0.932	0.973	0.514 ***
Canned fish	−0.212 (−0.419, −0.006)	−2.922	2.497	0.037
Shellfish	0.314 (0.073, 0.555)	−2.848	3.476	0.258 **
Fish	−0.024 (−0.204, 0.157)	−2.402	2.354	0.117
Eggs	0.070 (−0.089, 0.230)	−2.023	2.163	0.023
Vegetable oil	−0.069 (−0.259, 0.121)	−2.562	2.424	−0.132
Olive oil	−0.162 (−0.296, −0.027)	−1.924	1.600	0.078
Butter and margarine	0.086 (−0.025, 0.198)	−1.378	1.551	−0.005
Sweet and processed sugar	0.065 (−0.081, 0.210)	−1.859	1.988	0.080
Fruit juice	0.762 (0.432, 1.091)	−3.563	5.087	0.011
Coffee	0.013 (−0.197, 0.223)	−2.744	2.769	0.029
Tea	−0.232 (−0.548, 0.084)	−4.381	3.917	−0.040
Dipping sauces	−0.066 (−0.156, 0.024)	−1.242	1.110	−0.079
Soup	−0.011 (−0.208, 0.186)	−2.595	2.573	−0.054
Pizza	−0.231 (−0.479, 0.018)	−3.496	3.034	0.246
Wine	−0.076 (−0.253, 0.100)	−2.396	2.244	0.010
Alcoholic drink	−0.076 (−0.296, 0.143)	−2.963	2.811	0.054
Beer	−0.082 (−0.333, 0.169)	−3.379	3.216	−0.028
Salty snacks	−0.089 (−0.219, 0.042)	−1.803	1.625	−0.117
Fries	0.063 (−0.124, 0.250)	−2.393	2.520	−0.023
Fruit salad	0.093 (−0.128, 0.313)	−2.803	2.989	0.170 *

Abbreviations: 95% CI, 95% confidence interval; LOA, limits of agreement. *** *p* < 0.001; ** *p* < 0.01; * *p* < 0.05.

**Table 4 ijerph-16-01949-t004:** Mean agreement and limits of agreement between nutrient intakes estimated by the web-FFQ and the print-FFQ.

Nutrient Groups	Mean Agreement (95% CI)	Lower LOA	Upper LOA	Slope (*β*)
Total energy intake, kcal	−0.068 (−0.126, −0.010)	−0.835	0.699	0.135
SFA, mg	−0.072 (−0.137, −0.006)	−0.934	0.790	0.179 *
MUFA, mg	−0.239 (−0.354, −0.123)	−1.756	1.279	−0.427 ***
PUFA, mg	−0.092 (−0.160, −0.024)	−0.990	0.806	−0.020
Folate, µg	0.007 (−0.076, 0.090)	−1.077	1.091	0.058
Iron, mg	−0.059 (−0.120, 0.003)	−0.861	0.744	0.053
Calcium, mg	−0.060 (−0.066, −0,055)	−1.088	0.968	0.076
Magnesium, mg	−0.025 (−0.087, 0.036)	−0.835	0.784	0.080
Zinc, mg	−0.037 (−0.095, 0.020)	−0.800	0.725	0.145
Vitamin A, µg	−0.036 (−0.026, 0.051)	−1.172	1.101	0.034
Vitamin B1, mg	−0.027 (−0.067, 0.012)	−0.547	0.492	0.008
Vitamin B6, mg	−0.024 (−0.070, 0.023)	−0.636	0.589	0.056
Vitamin C, mg	0.179 (0.043, 0.316)	−1.613	1.972	0.029
Vitamin D, µg	−0.050 (−0.151, 0.052)	−1.383	1.283	0.163

Abbreviations: 95% CI, 95% confidence interval; LOA, limits of agreement. *** *p* < 0.001; * *p* < 0.05.

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
