# Peer review of "Comparison of Self-Administered Web-Based and Interviewer Printed Food Frequency Questionnaires for Dietary Assessment in Italian Adolescents"

_ijerph, 2019, doi:10.3390/ijerph16111949_

Round 1
Reviewer 1 Report
Revision of the manuscript entitled "Comparison of self-administered web-based and interviewer printed food frequency questionnaires for dietary assessment in Italian adolescents".
The article is well prepared in terms of structure and content.
The authors have presented a very good work.
Perhaps, the authors could extend the introduction by defining each of the study variables by means of new citations.
In line 39-40 of the introduction section, it is not necessary to put the DOI.
Another possible improvement would be to compare the results with studies carried out on the Mediterranean diet.
Likewise, in the section on materials and methods, the authors should put the ethics committee obtained for this research.
Author Response
Dear Editor,
thank you very much for considering our manuscript and for comments of independent Reviewers. We submit to your attention a revised version of the manuscript in which we have considered all comments. The following List of change and answers to comments of Reviewers addresses all changes made in the manuscript (red font).
List of change and answers to comments of Reviewers
Reviewer 1
Revision of the manuscript entitled "Comparison of self-administered web-based and interviewer printed food frequency questionnaires for dietary assessment in Italian adolescents".
The article is well prepared in terms of structure and content.
The authors have presented a very good work.
We are grateful to Reviewer 1 for his/her positive comments
Perhaps, the authors could extend the introduction by defining each of the study variables by means of new citations.
As suggested, we extended the Introduction section including some sentences about the rationale of our study and the need of innovative tools for dietary assessment in adolescents.
In line 39-40 of the introduction section, it is not necessary to put the DOI.
According to Reviewer 1 suggestion, we removed the DOI in lines 39-40
Another possible improvement would be to compare the results with studies carried out on the Mediterranean diet.
As suggested, we included a reference about the first web-based tool developed among Mediterranean adolescents (page 11 lines 3-4).
Likewise, in the section on materials and methods, the authors should put the ethics committee obtained for this research.
According to Reviewer 1 suggestion, we included the approval by the School Board of involved institutions in lines 29-30 (page 2).
Reviewer 2 Report
This is generally well written. I can understand the need to create a web-based FFQ, however, the correlations are quite poor. I am not a statistician, but it looks as though different correlations were tried to get some sort of result.
The questionnaire has 95 questions, which seems too long for young people. There is not description of the ethical considerations and how the adolescents were recruited, informed about the research. They were instructed on how to complete the questionnaire - but how would people without this instruction manage?
There is no discussion on why there may be differences in the correlations or consideration about future research. The poor correlations appear to be on healthy foods.
Specific points
Page 1 Line 25 - of should be changed to for
Line 37/38 - Need a reference to support that young people suffer chronic health conditions - how many? What type of conditions?
Page 2 Line 38 - Was there any bias due to presentation of the questionnaires?
Page 3 Line 28 - Was BMI self-reported or measured?
Page 9 Line 7 - blu needs an e added
Page 10 Line 8 - The weighted food diary is mentioned here but not in intro/method - how did the web FFQ measure up to the 7-day record?
Author Response
Dear Editor,
thank you very much for considering our manuscript and for comments of independent Reviewers. We submit to your attention a revised version of the manuscript in which we have considered all comments. The following List of change and answers to comments of Reviewers addresses all changes made in the manuscript (red font).
List of change and answers to comments of Reviewers
Reviewer 2
This is generally well written. I can understand the need to create a web-based FFQ, however, the correlations are quite poor. I am not a statistician, but it looks as though different correlations were tried to get some sort of result.
We are grateful to Reviewer 1 for his/her positive comments. We recognize that correlation coefficients were quite low but they were consistent with those reported by previous studies. Moreover, as reported in page 11 (lines 14-15), previous reviews reported a good correlation coefficient of 0.4.
The questionnaire has 95 questions, which seems too long for young people. There is not description of the ethical considerations and how the adolescents were recruited, informed about the research. They were instructed on how to complete the questionnaire - but how would people without this instruction manage?
In the present study, we adapted a self-administered web-based questionnaire from a validated interviewer printed questionnaire. The future aims of administering this tool will be the assessment of dietary habits – in terms of quality and quantity –, and specific dietary patterns. Accordingly, there is the need to assess dietary data about all the food items proposed.
According to Reviewer 2 suggestion, we included the approval by the School Board of involved institutions in lines 29-30 (page 2).
In this revised version of our manuscript, we better explained how students were enrolled and informed. Particularly, in future research, the application of the proposed web-based FFQ in large-scale studies will be accompanied by information sheets.
There is no discussion on why there may be differences in the correlations or consideration about future research. The poor correlations appear to be on healthy foods.
We recognize that correlation coefficients were quite low, but, as reported by previous reviews, a correlation coefficient of 0.4 can be considered good in this field of research.
Specific points
Page 1 Line 25 - of should be changed to for
As suggested, we changed “of” with “for”
Line 37/38 - Need a reference to support that young people suffer chronic health conditions - how many? What type of conditions?
We provided the requested references
Page 2 Line 38 - Was there any bias due to presentation of the questionnaires?
Limitation of the proposed tools, including those related to administration, are given in the discussion section
Page 3 Line 28 - Was BMI self-reported or measured?
As suggested, we specified that body weight and height were self-reported
Page 9 Line 7 - blu needs an e added
As suggested, we modified “blu” with “blue”
Page 10 Line 8 - The weighted food diary is mentioned here but not in intro/method - how did the web FFQ measure up to the 7-day record?
We apologize if this paragraph was confusing but this sentence was not related to our study. Indeed, we adapted a self-administered web-FFQ from a validated interviewer administered print-FFQ. Anyway, we revised this section to make it easier to understand.
Reviewer 3 Report
The authors presented the validity of the web-based FFQ, which might be useful for a survey. However, this reviewer has two major concerns.
Page10, Line 6-8. Criterion validation was not published in Ref. 16. “It has been previously validated against a 7-day weighted food record (WFR) (2014, unpublished observations).” This validation based on food record should be quoted in detail, and described which direction the bias observed in this study shift to, far away from the standards, or near to the standards.
Page 7, Table 2. Energy intake of the subjects was low. Medians are 1888–1927 (kcal). This may be quite low comparing the Italian reference of the same ages. Moreover, this value (15-18 years) is lower than the valued reported in the previous report, Ref 16 (14-17 years). Is the tool correctly applied to the field? First, discuss the underestimates. Second, do energy-adjustment and used adjusted values for statistical analysis.
Minor points
Units should be added in Table 2.
Author Response
Dear Editor,
thank you very much for considering our manuscript and for comments of independent Reviewers. We submit to your attention a revised version of the manuscript in which we have considered all comments. The following List of change and answers to comments of Reviewers addresses all changes made in the manuscript (red font).
List of change and answers to comments of Reviewers
Reviewer 3
The authors presented the validity of the web-based FFQ, which might be useful for a survey. However, this reviewer has two major concerns.
Page10, Line 6-8. Criterion validation was not published in Ref. 16. “It has been previously validated against a 7-day weighted food record (WFR) (2014, unpublished observations).” This validation based on food record should be quoted in detail, and described which direction the bias observed in this study shift to, far away from the standards, or near to the standards.
We are very grateful to Reviewer 3 for his/her comment. Accordingly, we have reported that findings from the validation study of the ASSO project have not yet been published.
Page 7, Table 2. Energy intake of the subjects was low. Medians are 1888–1927 (kcal). This may be quite low comparing the Italian reference of the same ages. Moreover, this value (15-18 years) is lower than the valued reported in the previous report, Ref 16 (14-17 years). Is the tool correctly applied to the field? First, discuss the underestimates. Second, do energy-adjustment and used adjusted values for statistical analysis.
We apologize for forgetting details about energy-adjustment (please see page 3 lines 3-4). Moreover, we discussed about low total energy intake observed in our population (page 11 lines 45-49).
Minor points
Units should be added in Table 2.
As suggested, we added units of measure in tables 2 and 4
Round 2
Reviewer 3 Report
The authors did not politely respond to the previous comments, and did not carefully plan the study protocol.
“A validated interviewer administered print FFQ” (Page 2, Line41-42) has 95 items, and the authors referred two articles (Refs. 22, and 23).
The first reference source is Ref.22, which used a 95 items FFQ for adult women aged 39 years, but is not a validation study. For validation, they referred Agodi’s report (European Journal of Clinical Nutrition (2011) 65, 1302–1308; doi:10.1038/ejcn.2011), which used only 46 items FFQ for child-bearing women, and showed validation only for folate.
The second reference source is Ref.23, which used a 95 items FFQ for women at cervical cancer screening. For validation, they referred the above Ref.22.
A FFQ that is known to be validated for adolescents about several nutrients cannot be used as a standard reference of this study. They cited their own articles (it is good) with not enough evidence (it is flaw).
Furthermore, the authors wrote, “To the best of our knowledge, only a web-based FFQ for dietary assessment in Italian adolescents has so far been developed within the Adolescents and Surveillance System for the Obesity prevention (ASSO) project [17]. However, findings from the validation study have not yet been published,” though the authors of Ref. 17 reported the validity using a 7-day weighted record for the ASSO project (Food & Nutrition Research 2015, 59: 26216 - http://dx.doi.org/10.3402/fnr.v59.26216). Interquartile range of energy intake using the web-based FFQ was 2339-4308 kcal. Why could the authors consider total energy intake <700 kcal or >4500 kcal plausible before analysis, on the faith just in the report for adult women?
Author Response
Dear Editor,
thank you very much for considering our manuscript and for comments of independent Reviewers. We submit to your attention a revised version of the manuscript in which we have considered all comments. The following List of change and answers to comments of Reviewers addresses all changes made in the manuscript (blue font).
Reviewer 3
The authors did not politely respond to the previous comments, and did not carefully plan the study protocol.
“A validated interviewer administered print FFQ” (Page 2, Line41-42) has 95 items, and the authors referred two articles (Refs. 22, and 23).
The first reference source is Ref.22, which used a 95 items FFQ for adult women aged 39 years, but is not a validation study. For validation, they referred Agodi’s report (European Journal of Clinical Nutrition (2011) 65, 1302–1308; doi:10.1038/ejcn.2011), which used only 46 items FFQ for child-bearing women, and showed validation only for folate.
The second reference source is Ref.23, which used a 95 items FFQ for women at cervical cancer screening. For validation, they referred the above Ref.22.
A FFQ that is known to be validated for adolescents about several nutrients cannot be used as a standard reference of this study. They cited their own articles (it is good) with not enough evidence (it is flaw).
We are sorry if our changes did not improve the manuscript properly and if provided references were confusing. In this revised version of our manuscript we did our best to better explain the study protocol and the development of FFQs. According to the Reviewer 3 suggestion, we clarified that the interviewer administered print-FFQ and the self-administered web-FFQ were adapted from a 95-item interviewer print-FFQ, which was previously used in the research on the effect of dietary habits on women health. The latter, in turn, was updated from a 46-item FFQ validated for the assessment of folate intake in Italian women of child-bearing age. Indeed, our aim was to develop and to compare FFQs for dietary assessment in adolescents. Both FFQs were adapted from a 95-item FFQ used in women from the Mediterranean region. We chose to adapt these tools for adolescents from a FFQ used in women since women, specifically mothers, have a crucial role in food choice by providing meals for their families.
Furthermore, the authors wrote, “To the best of our knowledge, only a web-based FFQ for dietary assessment in Italian adolescents has so far been developed within the Adolescents and Surveillance System for the Obesity prevention (ASSO) project [17]. However, findings from the validation study have not yet been published,” though the authors of Ref. 17 reported the validity using a 7-day weighted record for the ASSO project (Food & Nutrition Research 2015, 59: 26216 - http://dx.doi.org/10.3402/fnr.v59.26216).
We are grateful with Reviewer 3 for suggesting us the reference reporting the validation of the ASSO FFQ. Accordingly, we modified the text and included the suggested reference.
Interquartile range of energy intake using the web-based FFQ was 2339-4308 kcal. Why could the authors consider total energy intake <700 kcal or >4500 kcal plausible before analysis, on the faith just in the report for adult women?
As indicated by the Reviewer 3, in the study by Tabacchi and colleagues, interquartile range of energy intake using the web-based FFQ was 2339-4308 kcal, while it was 1658 – 2479 using a weighted food record. In our study, interquartile range of energy intake assessed by the web-FFQ was 1328-2618 and was similar to that observed by Tabacchi et al using the weighted food record. We considered implausible total energy intake as <700 kcal or >4500 kcal according to Rockett and colleagues (DOI: 10.1006/pmed.1997.0200). The authors defined outliers or implausible responses to the questionnaire as energy intake less than 500 kcal per day or greater than 5,000 kcal per day. In our study we used more stringent values to reduce the risk of potential measurement errors. According to Reviewer suggestion, we made it more clear in the discussion section and commented on differences in total energy intake between our and previous studies.